# A Roadmap of In Vitro Models in Osteoarthritis: A Focus on Their Biological Relevance in Regenerative Medicine

**DOI:** 10.3390/jcm10091920

**Published:** 2021-04-28

**Authors:** Isabella Bartolotti, Livia Roseti, Mauro Petretta, Brunella Grigolo, Giovanna Desando

**Affiliations:** 1Laboratorio RAMSES, IRCCS Istituto Ortopedico Rizzoli, via di Barbiano 1/10, 40136 Bologna, Italy; isabella.bartolotti@ior.it (I.B.); mauro.petretta@ior.it (M.P.); brunella.grigolo@ior.it (B.G.); 2RegenHu Company, Z.I Du Vivier 22, 1690 Villaz-St-Pierre, Switzerland

**Keywords:** osteoarthritis, in vitro inflammatory models, biomechanical models, microfluidics models, cartilage, synovium, 3D scaffolds, regenerative medicine

## Abstract

Osteoarthritis (OA) is a multifaceted musculoskeletal disorder, with a high prevalence worldwide. Articular cartilage and synovial membrane are among the main biological targets in the OA microenvironment. Gaining more knowledge on the accuracy of preclinical in vitro OA models could open innovative avenues in regenerative medicine to bridge major gaps, especially in translation from animals to humans. Our methodological approach entailed searches on Scopus, the Web of Science Core Collection, and EMBASE databases to select the most relevant preclinical in vitro models for studying OA. Predicting the biological response of regenerative strategies requires developing relevant preclinical models able to mimic the OA milieu influencing tissue responses and organ complexity. In this light, standard 2D culture models lack critical properties beyond cell biology, while animal models suffer from several limitations due to species differences. In the literature, most of the in vitro models only recapitulate a tissue compartment, by providing fragmented results. Biotechnological advances may enable scientists to generate new in vitro models that combine easy manipulation and organ complexity. Here, we review the state-of-the-art of preclinical in vitro models in OA and outline how the different preclinical systems (inflammatory/biomechanical/microfluidic models) may be valid tools in regenerative medicine, describing their pros and cons. We then discuss the prospects of specific and combinatorial models to predict biological responses following regenerative approaches focusing on mesenchymal stromal cells (MSCs)-based therapies to reduce animal testing.

## 1. Introduction

Osteoarthritis (OA) is one of the most globally spread joint disorders with a strong impact on patients’ daily lives [1]. It affects approximately 630 million people worldwide with a different incidence rate in men and women, and an estimated increase by 2050 is forecast [2,3]. Age, sex, genetic profile, lifestyle, and obesity are among the leading risk causes of OA [4]. Despite the number of studies, many gaps exist in understanding thoroughly biological mechanisms beyond OA pathogenesis [5]. Several preclinical in vitro and in vivo studies elucidated some cellular and molecular mechanisms and provided first evidence on potential new treatments’ regenerative potential [6,7,8,9]. In particular, it was shown that the OA milieu displays a plethora of degenerative and phlogistic processes, which evolve at the end-stage in a joint’s destruction [10,11].

The motivation for developing versatile in vitro OA models, simulating the disease complexity, is twofold: first, to achieve a better grasp on disease understanding; and second, to look for novel therapeutic options for proof-of-concept studies [6,7,12]. Considerable attention is crucial in selecting the proper model to mimic OA conditions, depending on the biological issues under investigation. Two-dimensional culture models enable testing of the effect of treatments only at the cellular level, as they lack typical 3D microstructure and the crosstalk with the surrounding cell types. Using both 2D and 3D coculture systems offers the great advantage of building up a realistic microenvironment to study cell–cell interactions in the joint. Moreover, the possibility of recapitulating physical, chemical, and biomechanical OA stimuli in combinatorial models allows testing of different strategies through a multidisciplinary approach [13,14,15,16,17,18,19,20,21,22]. Exploiting the potential of new in vitro OA models might increase the state of the art of signaling pathways involved in catabolic, inflammatory, and oxidative stress responses. This review provides an overview of the old-fashioned and innovative in vitro models and offers a glimpse of the limits and strong points of research and therapeutic tools focusing on mesenchymal stromal cells (MSCs)-based therapies.

## 2. Osteoarthritis: Catabolic and Inflammatory Mechanisms and Therapeutic Modalities

### 2.1. Organ Complexity in Osteoarthritis: Pathophysiology

OA was considered for a long time as a “wear and tear” disorder caused by articular cartilage degradation with direct consequences on the subchondral bone [10,11]. In the last decade, the above definition of OA has been changed to “whole joint” disorder, as it shows a direct contribution not only of articular cartilage but also of the synovial membrane, subchondral bone, and meniscus in triggering OA onset and progression [10,11,23,24]. Intrinsic or extrinsic stimuli can lead to impairment of the articular cartilage resulting in fibrillation/erosion of the superficial/mid or deep zones. These structural changes determine chondrocyte “activation” and promote the synthesis of extracellular matrices (ECM) degradative enzymes such as matrix metalloproteinases (MMPs) and aggrecanases (ADAMTS). MMP-13 and ADAMTS-5 are among the major catabolic enzymes in mediating collagen and glycosaminoglycans (GAGs) breakdown in articular cartilage with consequent joint destruction [1]. In turn, MMPs and ADAMTS foster the release of inflammatory factors such as interleukin 1-beta (IL-1β), tumor necrosis factor (TNF-α), cyclooxygenase 2 (COX2), IL-6, INF-λ, [25,26]. These mediators are modulated by several signaling pathways including the nuclear factor kappa-light-chain-enhancer of activated B cells (NFKB), NOTCH [27], and epigenetic pathways with marked changes of the miRNome [28]. Beyond chondrocytes, immune cells and synoviocytes fuel this vicious circle, triggering the release of more cytokines [29,30]. Among immune cells, macrophages with a pro-inflammatory phenotype (M1 subset) release a broad spectrum of phlogistic and immune mediators into the synovial cavity, contributing to cartilage loss and osteoclastogenesis [31,32,33,34]. Following inflammatory stimuli, chondrocytes produce abnormal levels of reactive oxygen species (ROS), due to mitochondrial dysfunction, by boosting cellular signaling and matrix catabolism [35]. Typical hallmarks of OA progression comprise joint space narrowing with cartilage loss, tidemark duplication, cyst formation, thickening of subchondral bone, and inflammation of the synovial membrane [36,37]. Stiffness, pain, and impaired movement result from these structural changes [38].

### 2.2. Therapeutic Options for Osteoarthritis Treatment 

Although different therapeutic options for OA care are present, they are tailored to relieve pain, and improve joint mobility and function without eradicating the disease. These strategies include non-pharmacologic treatments, intra-articular therapies, biological alternatives, and when the patient’s quality of life is completely compromised, the last choice is the surgical procedure with total joint replacement [39,40,41]. As for pharmacologic treatments, non-steroid anti-inflammatory drugs (NSAIDs) are among the first-line treatment modalities to counteract inflammatory-related symptoms through the inhibition of COX-2, which mediates the conversion of arachidonic acid to prostaglandin endoperoxide H2 (PGE2) [42]. However, NSAIDs exert several side effects on cartilage metabolism with important implications for its regenerative ability [43]. Beyond NSAIDs, further therapeutic alternatives ranging from physical strategies (physical activity, laser, etc.), intra-articular injections of several compounds (corticosteroids, hyaluronic acid (HA), and platelet-rich plasma (PRP), have been proposed as minimally invasive treatments to counteract joint destruction [44,45,46]. Despite their promising effects, their administration is strongly suggested in patients with early OA and not in moderate and severe stages [47]. Oral supplementation with different dietary supplements (chondroitin sulfate, glucosamine, etc.) represents a further therapeutic option for treating early OA due to their anti-inflammatory action [48,49]. In the last decade, great efforts are addressed in developing more effective alternatives with long-term effects. Nonoperative interventions including the use of cell-based therapies and tissue engineering approaches are highly increased. First attempts focused on the use of autologous chondrocytes to reestablish cartilage integrity; however, further cell sources have been considered due to the loss of phenotypic stability of chondrocyte and morbidity near tissue harvesting [50]. The feasibility of using MSCs from bone marrow and other tissue sites, based on their ability to influence and regulate different stages of tissue repair, is a great challenge among clinicians [51,52]. MSCs can be present as expanded cells, following tissue collection and isolation, or as a heterogeneous population after concentration processes with different devices [53,54,55,56]. Expanded MSCs and concentrated progenitors undergo different regulatory compliance, as the first approach foresees cell manipulation in GMP facilities and two-step procedures, whereas the second one requires minimal manipulation and occurs through a one-step procedure directly in the operating room [57]. Although cell-based strategies are gaining great attention, several outstanding questions—including the selection of cell source, the choice of autologous versus heterologous cells, the definition of dose, the number and timing of treatments, and their migratory profile—remain. Therefore, scientists are attempting to find valuable and versatile preclinical models, simulating the multifaceted and complexity of this illness, to unravel these aspects.

## 3. Pros and Cons of In Vitro Inflammatory Models: Their Potential in Regenerative Medicine 

### 3.1. Biological Relevance of Inflammatory In Vitro Models for Investigating OA

The inflammatory environment is a typical hallmark of OA patients, which triggers several catabolic processes with loss of tissue integrity and consequent damages to the articular joint [58,59]. The synovial membrane is among the main tissues implicated in OA-related inflammation displaying marked hyperplasia of the synovial lining and infiltration of inflammatory cells, comprising macrophages and, to a lesser extent, B, T, and NK cells [60,61]. Beyond classical inflammation due to physical traumatic events, recently great attention has been paid to meta inflammation promoted by excessive metabolites and nutrients such as lipids—such as occurs in obese patients [62]. Regardless of the cause, the inflammatory milieu has important consequences on several biological mechanisms implicated in the modulation of pain [63], catabolic processes [64], and stromal cell niche [64]. In particular, the stromal cell niche plays a pivotal role in ensuring an adequate response following tissue damages; its impairment can alter the tissue repair machinery. Many studies focused on elucidating the impact of inflammation on (1) cell survival; (2) cell migration; and (3) MSCs functions. Employing exogenous cytokines via in vitro systems is one of the most commonly used methods to induce inflammation [6]; therefore, the inflammatory model is also known as the “cytokine-based model”. In vitro, inflammatory models have the important role of elucidating biological processes in OA cartilage and synovium and of testing alternative models of therapeutic interventions targeting inflammation. When selecting the inflammatory model, two main crucial aspects have to be considered. First, selecting culture models (cell lines versus primary cell cultures, isolation procedures for primary cell cultures, 2D/3D, or cocultures) is a crucial parameter with an enormous impact on tissue response. Second, selecting inflammatory stimuli is another critical parameter to select before starting the experimentation through an accurate evaluation of their pros and cons. The adoption of validated protocols for isolating and expanding primary cells covers great relevance to optimize cell survival and to avoid the loss of their biological properties [65,66,67,68,69]. Along this line, Manferdini et al. have shown that the isolation procedure of cells in the synovial membrane has a great impact on the phenotypical and functional properties of synovial fibroblasts/macrophages. In particular, the mechanical procedure preserves better than the enzymatic one in the in vivo heterogeneous population of the synovial populations [66]. Moreover, synovial cells cultured at passage 1 preserve both macrophages and fibroblasts, whilst at passage 5 only fibroblasts are preserved [67]. These results open an interesting scenario in which we are able to mimic low and moderate stages of OA synovitis through different culture passages. Most of the in vitro studies are carried out in specimens from OA patients who underwent joint prosthesis and who likely exhibit a basal impairment in tissue homeostasis. The inability of obtaining human healthy tissues devoid of inflammatory stimuli from surgical samples is a big issue to consider during the experimental design. In this context, several companies offer quality-controlled healthy cells from various tissue sources to help investigators to ensure control groups. 

### 3.2. 2D and Coculture Systems in Cartilage and Synovium

The first culture models to study OA used 2D cultures of chondrocytes, focusing on biological aspects related to cartilage changes but without considering the joint complexity [70,71,72,73,74,75,76]. To overcome these limitations, scientists extended their research to other joint tissues such as the synovial membrane, which is composed of a heterogeneous cell population of fibroblasts and macrophages [77]. As for the synovium, the first preclinical studies encompassed testing 2D cultures of synovial fibroblast and cell lines such as RAW 264.7, K4IM; however, their major limitations included the lack of analyses on the synovial macrophages and extracellular matrix [78,79]. Indeed, the use of 2D culture models is more suitable for synoviocytes than chondrocytes, as this latter does not retain a stable phenotype over culture passages and limit the number of experiments [6,80]. Notably, Hung, C.T. and his coworkers gave evidence of the biological relevance of lavage and synovectomy for OA through cocultures of synovial fibroblasts with cartilage and latex particles with IL1α and TNFα [78]. The feasibility of in vitro modeling the complex cross-talk between cartilage and the synovial membrane is challenging as the synovium and especially macrophages play a pivotal role in mediating cartilage matrix breakdown [24,81]. The first attempts of cocultures provided insightful indications of OA-related degeneration processes. These experiments gave evidence about the release of many inflammatory cytokines not only from the synovial membrane but also from the articular cartilage, by feeding a vicious inflammatory circuit [82,83,84]. For all these aspects, we can consider coculture systems a more complex model than monolayer cultures to recapitulate the inflammatory network. Unfortunately, results from both 2D monolayer cultures and cocultures can undergo differences due to donor-related variability, which has to be considered when interpreting the results. Indeed, the preliminary selection of cartilage and synovium with similar macroscopic features before cell isolation can help to limit the variability. When referring to cocultures, examining the crosstalk of chondrocytes with macrophages/T cells is demanding to get closer to assessing immunomodulation. Lohan, P. et al. proved reduced MHCII and TNFα expression in cocultures of rat chondrocytes and allogeneic macrophages. Moreover, they showed a drop in T-cell growth in cocultures of chondrocytes with allogeneic T lymphocytes driven by nitric oxide [85]. In general, these systems provided insightful evidence of the impact of synoviocytes on cartilage features and function under inflammatory stimuli [84]. Despite the biological value and insightful perspective from 2D models, they display several drawbacks since they do not mirror the complexity of the OA microenvironment and 3D architecture by limiting mechanistic insights (Table 1).

### 3.3. 3D and Coculture Systems in Cartilage and Synovium

The plethora of 3D cartilage models in OA include the use of (i) alginate beads [85]; (ii) pellet cultures [86,87,88,89]; (iii) the combination/embedding of chondrocytes with different scaffolds (e.g., hydrogel, polymeric, fiber/mesh scaffolds, etc.) [90,91,92,93,94,95], (iv) tissue explants [96,97]. Using 3D systems is beneficial for maintaining the phenotypic and functional features of articular chondrocytes, which would be lost following 2D cultures. The pellet culture represents one of the first and simplest 3D cartilage models. This model promotes a better chondrogenic differentiation of OA chondrocytes than MSCs cultured with recombinant growth factors such as TGF-β. However, it shows the limit of presenting jeopardizing areas of chondrogenesis and necrotic areas in the central region of the pellet due to improper nutrient diffusion [98]. Another option, largely employed for studying articular cartilage, is the use of osteochondral explants, since they recapitulate the 3D ECM-cell and cell–cell interactions of the osteochondral unit; thus, offering the advantage of keeping articular cells in their native milieu with the nourishment from the underlying subchondral bone [96,97]. Unfortunately, explants can preserve natural properties for a short time and only a few repeats from the same source are available; thus, making difficult the standardization and interpretation of results [99]. Further approaches for generating 3D models for articular cartilage arise from regenerative medicine. These strategies comprise a wide plethora of biomimetic materials (hyaluronic acid, different collagens, etc.) reflecting the native articular cartilage. Most of these biomaterials provided interesting and promising results, as they allow the maintenance of the chondrocyte phenotype [100,101]. Similarly to articular cartilage, alternatives for mimicking the 3D structure of the synovial membrane can include the use of (i) tissue explants; and (ii) embedding of cells within scaffolds. Recently, authors proposed the synovium joint capsule (SJC), the tissue enclosing the joint, as a potential tool to in vitro model OA for studying aggrecan proteolysis. Evaluating SJC is mainly due to the presence of the synovial membrane in its inner structure, modulating the volume and composition of SF [102]. Unfortunately, these tissue explants cannot mimic hyperplasia and fibrosis in a standardized way due to the presence of different anatomical regions [103,104]. Vankemmelbeke, M.N. et al. showed that synovium and capsular-derived tissue contribute to increasing soluble aggrecanase activity after their coculture with articular cartilage [105].The expression of ADAMTS4, ADAMTS5, and their proteolytic products in the SF from OA patients contributed to corroborating the relevant biological value of synovial membrane in inflammation [106]. Notably, Stefani, R.M. et al. developed a tissue-engineered synovium model by combining bovine synovial fibroblasts and macrophage-like synoviocytes encapsulated in Matrigel^®^. They reported Matrigel^®^ as a valid tool for exploring the influence of inflammatory processes by synovial cells on the articular joint [78]. Along this line, Samavedi et al. proposed in vitro models of early and late OA, presenting cocultures of normal and OA chondrocytes and macrophages with an inflammatory phenotype (M1), both set in poly- (ethylene glycol) diacrylate hydrogel [107]. To expand the study on the main targeted OA tissues, Haltmayer, E. and her group developed a multimodal approach of chemical and mechanical stimuli. They set up cultures of osteochondral–synovial membrane explants supplemented with IL1β and TNFα, with a partial-thickness cartilage defect. This model allowed the shift of synovial macrophages towards M1 phenotype by upregulating MMPs and ADAMTS5 [108]. The feasibility to mimic OA through combinatorial models is challenging as it allows the simultaneous study of different biological stimuli, which influence cell response (Table 1).

### 3.4. Biological Sources of Inflammatory Stimuli 

To date, it is possible to distinguish different biological sources of inflammatory stimuli, which include the use of (i) cytokines; (ii) synovial fluid from OA patients; (iii) activated macrophages, and (iv) mediums conditioned by macrophages (MCM) or from OA cells under the form of extracellular vesicles (EVs) [6,109,110]. IL1β and TNFα are among the two most common pro-inflammatory cytokines studied in OA [111,112,113]. IL1β increases in the early and late stages of the disease and TNFα in the OA onset by launching signaling pathways involved in tissue destruction [70,71,72,73,74,75,76]. Beyond catabolic processes, IL1β inhibits the synthesis of type II collagen and proteoglycans. Moreover, it induces apoptosis and oxidative stress in chondrocytes by increasing inducible nitric oxide synthase (iNOS) and ROS [114,115]. For all these reasons, these inflammatory models provided important insights in elucidating several mediators involved in tissue destruction, such as MMPs and ADAMTS. Beyond the use of IL1β and TNFα [111,112,113], scientists developed a combined model of hypertrophy and inflammation using cultures of OA chondrocytes supplemented with transforming growth factor-beta 1 (TGFβ1), a pleiotropic cytokine with a functional role in both healthy and OA joint [116,117,118]. TGF-β is a well-known stimulator of synovial inflammation and hyperplasia in the OA setting and it is present for a short period and at low levels after joint loading in health conditions [116,117,118]. Using single or combined doses of cytokines partially reflects the in vivo microenvironment, which shows a complex biological scenario with a high risk of underestimating possible biological reactions due to improper selection of concentration or timing of stimulation. The concentrations of cytokines used in vitro for recapitulating inflammation are much lower than those detected in the synovial fluid of OA patients [6]. Synovial fluid from OA patients offers a more complex cytokine network than using exogenous cytokine stimulation. Scientists observed that stimulating articular chondrocytes with the synovial fluid from OA patients promotes an inflammatory phenotype (release of IL-6, IL-8, VEGF, and MCP-1) [119]. As for fibroblast-like-synoviocytes, their stimulation with the synovial fluid from patients with early OA enhanced their response to TRL-2 and TRL-4 ligands [120]. Although the synovial fluid is a valid tool for recapitulating OA in both chondrocytes and synoviocytes, it allows a reduced number of experimental studies because of its biological variability [16,121]. As for activated macrophages, Samavedi et al. gave evidence that their crosstalk with healthy chondrocytes embedded in a hydrogel promoted the release of key inflammatory mediators by mimicking the early stage of OA. Conversely, the crosstalk between OA chondrocytes and activated macrophages may allow the modeling of severe stages of OA [107]. The main limitation of this study was the use of cells deriving from two different species, mice and humans, with important implications for the final biological responses. Recently, increasing attention is on the use of the MCM, as macrophages are among the major producers of inflammatory cytokines in OA [122,123,124]. When referring to MCM, it is extremely important to report the macrophage subset. Utomo, L. et al. proved that the MCMs from M1 macrophages (INFγ + TNFα) have a great effect on OA cartilage explant, whereas M2 macrophages (IL4 and IL-10) may not inhibit the effects of M1 [123,124]. Comparisons between the use of direct cytokine stimuli (IL1β + TNFα) and MCM gave evidence that the latter promotes—in the best way—immune-related OA changes. Moreover, MCM treatment exerts multiple effects not only on anabolic and catabolic processes but also on chondrocyte hypertrophy and apoptosis [125]. These results would suggest how this kind of stimuli can be applied in elucidating processes beyond OA onset and progression and also modes of intervention. Recently, the use of EVs has been gaining great attention in the field of OA [126,127]. EVs act on synovial fibroblasts modulating the release of inflammatory cytokines and chemokines [128]. Several lab scientists tested the effect of exosomes from IL1β stimulated synovial fibroblasts on chondrocytes. Interestingly, Kato, T. et al. proved that EVs upregulate MMP13, ADAMTS5, and downregulate collagen type II and aggrecan in chondrocytes [129] (Table 1). 

**Table 1 jcm-10-01920-t001:** List of inflammatory Osteoarthritis (OA) models with specific details of the nature of the inflammatory stimuli, the culture systems (2D/3D/cocultures), the main findings, and the bibliographic reference.

Inflammatory Stimuli	Culture Models	Main Results	Ref.
TNF-α	Monolayer culture of human articular chondrocytes	Activation of ERK, JNK, p-38, and NF-Kβ signaling pathway.	[73]
	Monolayer culture of human OA chondrocytes	Marked expression of MMPs and inhibition of anabolic molecules, such as type II collagen and proteoglycans.	[112]
	Monolayer culture of human OA chondrocytes	Induction of chondrocyte apoptosis.	[115]
IL1β	Monolayer culture of human chondrocytes	Induction of MMP-1, -3 and -13 and ADAMTS; Inhibition of type II collagen and proteoglycans.	[72,73,109,112]
	Monolayer culture of human articular chondrocytes	Activation of ERK, JNK, p-38, and NF-Kβ signaling pathway.	[73]
	Equine cartilage explants	Increase in gene expression and protein release of cytokines, chemokines, and MMPs.	[74]
	Monolayer culture of human chondrocytes	Induction of inflammation mediated by IL-6 synthesis with no modulation of PGE2.	[110]
	Monolayer culture of bovine chondrocytes	Deregulation of the enzymatic antioxidant defences in chondrocytes with mitochondria disfunction.	[114]
IL1β + TNFα	Equine articular chondrocytes	Noticeable alterations of various matrix-related gene expression involved in cartilage breakdown.	[71]
	Coculture of osteochondral and synovial explants	- Induction of inflammatory and catabolic processes; - Shift towards the pro-inflammatory M1 synovial macrophages.	[108]
	2D culture of human OA chondrocytes	Induction of chondrocyte apoptosis.	[115]
TGFβ	Monolayer culture of chondrocytes	Induction of hypertrophic features including marked levels of RUNX2, type X collagen, MMP13, VEGF, and activation of the canonical Wnt pathway.	[116,117,118]
TNFα/TGFβ	3D synovial micromass model (synovial cell suspension and primary fibroblasts (FLS) and CD14+ monocytes)	3D synovial micromasses, following the inflammatory stimuli, expressed pro-inflammatory cytokine, hyperplasia, and fibrosis-like changes	[104]
Macrophages	2D coculture of rat chondrocytes with allogeneic macrophages	Modulation of pro-inflammatory macrophage activity through the reduction of MHCII and TNF-α	[85]
	3D coculture of normal chondrocytes (NC) + activated macrophages (M1) in PEDGA hydrogel	Early OA induction: increase of MMP-1 and MMP-3 mRNA by cartilage explants and pro-inflammatory cytokines (IL-1β, TNF-α, IL-8, IFN-γ, and MCP-1).	[107]
	3D coculture of OA chondrocytes + M1 within PEDGA hydrogel	Late OA induction: high expression of MMP-1, MMP-3, MMP-9, MMP-13, IL-1β, TNF-α, IL-6, IL-8, INF-γ, VEGF-A	[107]
	OA cartilage explants + MCM from M1 macrophages	Induction of several OA features typically observed following cytokine stimulation.	[122]
	3D culture of primary chondrocytes in silk scaffolds + MCM	- Promotion of type X collagen and chondrocyte apoptosis; - Induction of typical OA catabolic and inflammatory processes.	[123]
Synovial Fluid (SF)	Primary human chondrocytes	SF activates pro-inflammatory cytokines: IL6, IL8, VEGF, MCP1	[119]
Synovial Joint capsule (SJC)	Bovine articular cartilage explants	Induction of proteoglycan-degradation and MMPs	[102]
Exosomes from IL-1β stimulated synovial fibroblasts	2D culture of human chondrocytes/mouse femoral head cartilage explants	Induction of OA-like changes through the secretion of MMPs, VEGF and inflammatory cytokines	[129]

### 3.5. Perspectives of Inflammatory OA Models in Regenerative Medicine: A Focus on MSCs

Targeting inflammatory processes represents a challenging treatment option to counteract OA progression. Countless applications have been developed, ranging from NSAIDs and corticosteroid injections to several biological treatments targeting macrophage-produced cytokines; however, they showed several side effects [52,60]. Recently, the use of cell-based approaches based on MSCs from different biological sources gained great attention thanks to their differentiation and paracrine activities through the release of immune-suppressive, anti-inflammatory, anti-apoptotic, and regenerative mediators [52]. The feasibility to apply in vitro inflammatory models has been considered for testing the functional properties of progenitor cells in OA [52]. When evaluating the therapeutic potential of MSCs, three key aspects have to be considered: (i) their migratory pattern in the injured area; (ii) their differentiation potential; and (iii) their paracrine activities [130].

#### 3.5.1. Inflammatory Models for Examining the Migration Pattern of MSCs 

Identifying mechanisms promoting MSCs migration is crucial to elucidate their chemotactic and regenerative properties on the target tissues. Several studies tracking cell biodistribution through in vivo OA models reported cell migration, especially near the inflammatory areas of the synovial membrane following intra-articular administration [131,132,133,134]. To gain new insights on the biological mechanisms driving GMP-adipose-derived mesenchymal stromal cells (ASCs) in the OA milieu, Manferdini, C. et al. carried out an in vitro study using synovial fluid from OA patients or conditioned medium from OA synoviocytes as inflammatory stimuli. They demonstrated that the established in vitro inflammatory environment modulates the migration and cytokine receptor expression of GMP-cultured ASCs with important implications on their efficacy by modulating the CXCL-10/IP10/CXCR3 axis [135]. Recently, scientists provided new insights on the effect of EVs from MSCs on cell migration, underlining a dose-dependent effect [136]. Although these in vitro studies elucidate pathways beyond cell migration, it is crucial to consider that CXCR4, the main surface marker involved in cell homing, is decreased over culture passages in MSCs [137,138]. Its decrease influences MSCs’ ability to answer to the homing signals from injured joint tissues.

#### 3.5.2. Inflammatory Models for Examining the Chondrogenic Commitment of MSCs 

The inflammatory milieu contributes to cartilage destruction. Because of its avascular nature, cartilage injuries are difficult to repair. Exploiting the multipotential ability of MSCs has been considered among the therapeutic options for treating articular cartilage. In this light, several studies were tailored in testing the impact of the inflammatory milieu on the differentiation ability of MSCs towards chondrogenic lineage. Physiological chondrogenesis is driven by specific transcription factors and cytokines (TGF-β, IGF-1, BMP, FGF), which promote the transcriptional activation of chondrogenic genes via the type I receptor (ALK5) and phosphorylation of SMAD2/3. However, the main limitation of the chondrogenic differentiation is its possible evolution towards hypertrophic phenotypes through the binding to ALK1 and phosphorylation of SMAD1/5/8 [139]. In general, these studies gave evidence that inflammatory stimuli such as IL-1β and TNF-α inhibit chondrogenic differentiation via the TGF-β/SMAD signaling pathway [140,141].

#### 3.5.3. Inflammatory Models for Examining the Paracrine Activities of MSCs 

Much evidence on the immune-mediated responses from MSCs comes from in vitro inflammatory models. It is well-known that MSCs become activated following exposure to the inflammatory environment with important perspectives in regenerative medicine [142]. Cocultures of bone marrow-derived MSCs and monocytes from the peripheral blood under inflammatory stimuli promote the polarization of this latter towards the M2 macrophage subset through the release of soluble factors (PGE2, IDO, etc.) [143,144]. Similar findings were observed for ASCs, which can switch off the activated M1 subset [145]. Taken altogether, these findings highlighted the importance of cell-to-cell contact between MSCs and joint tissues. To gain more insights on the impact of the inflammatory milieu on the regenerative potential of MSCs, several in vitro studies have been carried out. To this end, it has been demonstrated that 3D cocultures of MSCs with primary chondrocytes promote a higher chondrocyte proliferation than 2D models. Fibroblast growth factor 1 (FGF-1) was identified among the main mediators involved in the proliferation of primary chondrocytes [146]. Along this line, Maumus, M. et al. demonstrated the anti-fibrotic effect of MSCs from bone and adipose tissues through the set-up of coculture systems with OA chondrocytes. They noticed that MSCs can downregulate hypertrophic and fibrotic markers via the hepatocyte growth factor (HGF); thus, allowing the maintenance of the chondrocyte phenotype [147]. As for ASCs, several studies were carried out to explore whether there were biological differences related to their anatomic site. To this end, some scientists investigated the anti-inflammatory behavior of three fat sources: infrapatellar Hoffa fat, subcutaneous hip fat, and abdominal fat. Finally, they demonstrated that all three sources of GMP–ASCs were able to downmodulate IL-1β, IL-6, and CXCL8/IL-8 via the COX-2/PGE2 pathway using cocultures between OA chondrocytes/synoviocytes and ASCs [148]. Taken altogether, these studies demonstrated that the coculture systems are valid tools to examine the anti-inflammatory potential of MSCs. Recently, many studies are focusing on elucidating the effect of EVs from MSCs, consisting of proteins, lipids, nucleic acid, and other components, on tissue regeneration. MSCs produce a wide array of EVs, which play a pivotal role in cell-to-cell communication with important implications for tissue repair [149,150,151]. Coculturing 2D OA chondrocytes following IL-1β stimulation with EVs from bone marrow-derived MSCs has a chondroprotective effect as it upregulates COL2A1 protein and downregulates MMP-3 and ADAMTS5 [149]. Similar findings have been obtained by coculturing OA chondrocytes with EVs from ASCs, where not only a decrease of MMPs but also of some inflammatory mediators such as TNF-α, IL-6, PGE2, and NO were noticed [152]. Recently, Cavallo, C. et al. have investigated the EVs from ASCs not only on OA chondrocytes but also on OA synoviocytes under culture conditions with IL-1β. EVs from ASCs were able to inhibit IL-1 inflammatory effects through the NF-Kβ pathway in both chondrocytes and synoviocytes, with stronger effects in the latter [153]. When using exosomes, it is important to consider proper isolation methods, as ultracentrifugation procedures via external forces can damage their structure. In summation, these inflammatory in vitro models represent valid tools for screening the regenerative potential of MSCs.

## 4. Pros and Cons of In Vitro Biomechanical Models: Their Potential in Regenerative Medicine 

### 4.1. Biological Relevance of Biomechanical In Vitro Models for Investigating OA

Biomechanical models, also known as loading-based OA models, are addressed to develop OA-like changes in joint tissues by introducing supra-physiological mechanical stresses [154,155,156]. Employing these models envisages increasing the comprehension of (i) the influence of biomechanics on joint metabolism, and (ii) the impact of supra-physiological mechanical forces on tissue regeneration. Intrinsic and extrinsic mechanical forces influence musculoskeletal tissue via mechano-transduction; its impairment often leads to traumatic events that evolve in the onset of OA. Bidirectional interaction between cells and ECM promotes cellular development, physiology, and adaptive remodeling [157,158,159,160]. During direct mechano-transduction, physical forces act on integrins linked to the cell nucleus with the cytoskeletal proteins. Finally, these forces determine gene expression changes in chromatin with subsequent effects on cell shape, migration, and differentiation [161,162,163,164,165]. Indirect mechano-transduction occurs mainly in two ways through the activation of integrin-mediated pathways or mechano-sensitive ion channels. Load-based models can envisage the use of 2D or 3D culture systems (with/without biomaterials) [6]. Indeed, 3D structures ensure a better cell response to biophysical factors than 2D models as they promote more complex and dynamic changes [166]. Using tissue explants has the benefit of keeping cells into their native cell-matrix interactions, whereas cells can be embedded or loaded in scaffolds based on their nature [6,22]. Scaffold architecture is a critical parameter that modulates the number of applied forces received by cells. Embedding cell in hydrogels ensures a more homogeneous diffusion of mechanical forces than porous scaffolds [167,168,169,170,171,172]. The number of modalities whereby scientists can mechanically stimulate cells is variable and the available devices can be categorized based on the primary type of stress they induce (compression, shear forces, etc.) [167]. Combining these models with perfusion bioreactors, spinning flasks, rotating wall vessels, and microfluidics may be valid research methods to mimic the loads and motion patterns of tissues with important insights in regenerative medicine [6,173,174,175].

### 4.2. Evaluating Biomechanical OA Models in Osteochondral Tissue 

Articular cartilage has a complex structure organized in different layers, which display different proteoglycan/collagen content, cell alignment, morphology, and density. This architecture results from the dynamic processes this tissue undergoes (e.g., compression and shear stress) [171]. During the joint loading, cartilage ECM holds water from the synovial fluid thanks to the negative charges of proteoglycans leading to an increase of hydrostatic pressure (HP). The distribution of collagen fibers in the different layers of cartilage ECM has a pivotal role in preventing tissue swelling. HP from 7 to 10 MPa is beneficial for promoting ECM synthesis in terms of an increase of type II collagen and aggrecan. In this context, uniaxial and multiaxial loading systems represent valid models as they allow the upregulation of chondrogenesis-related genes [172]. Conversely, supraphysiological forces such as injuries and excessive loads cause several articular changes responsible for joint destruction [176]. “Drop towers” are among the most commonly used loading models and are useful for scientists to solve biological questions related to post-traumatic OA. This model exerts a single impact load on tissue explants from a certain height with either static or cyclic modalities. Applying this model, Torzilli, P.A. et al. showed a direct relationship between increasing impact stress and cartilage injuries. This group showed that threshold stress of 15–20 MPa induces cell death, proteoglycan depletion, and disruption of collagen fibers [177]. It is well-known that articular cartilage can answer mechanical stimuli through the release of intracellular calcium ions (Ca^2+^), as it displays ion channels on its surface. It is possible to distinguish three types of ion channels on cell membrane: TRPV4 (responsive to osmotic and mechanical stress), T-type VDCCs (sensitive to electrical stimuli), and mechanical sensitive ion channels (sensitive to mechanical loading) [171]. There is growing evidence on the relationship between intracellular Ca^2+^, cartilage changes, and inflammation [178,179]. Along this way, Guilak and his coworkers evaluated the impact of IL1β on porcine articular chondrocytes following mechanical loading. They proved that IL1 β interferes with the adaptative responses to mechanical loading (osmotic stimulation) by altering the Ca^2+^ response through the F-actin remodeling mediated by small Rho GTPases [178]. Among mechanical forces, compression is one of the most studied, as it affects cartilage health with variable force in anatomical sites according to weight, muscular tension, and physical activity [180]. Tissue explants and scaffolds/hydrogels are suitable tools to test tissue metabolism in dynamic conditions considering different compressive loading frequencies (0.001–5 Hz) [172]. Supraphysiological loading determines cell death, proteoglycans loss, and collagen network disruption [181,182,183,184,185]. Notably, Madej, W. et al. demonstrated that excessive mechanical compression can activate the SMAD 2/3P signaling pathway with a chondroprotective role [186]. In this context, Dolzani, P. et al. investigated the impact of mechanical stimuli on OA cartilage explants in different experimental conditions. Findings from this study gave evidence that this mechanical force influences the effect of the inflammatory microenvironment by modulating cartilage mediators involved in tissue metabolism [187]. Recently, Nakamura, F. et al. have established the role of angiotensin II type 1 receptor (AT1R) as a mechano-sensor involved in OA progression. To this end, they encapsulated bovine chondrocytes in agarose scaffolds under cyclic compression culture conditions with and without an Ang II receptor blocker. Findings from this study showed an increased gene expression of collagen type X and Runx2, typical hypertrophic markers present in OA. Inserting the Ag II receptor blocker reversed the hypertrophic phenotype [168]. Similarly, to cartilage, osteocytes entrapped in the ECM are the primary sensors of mechanical forces, by regulating the local mineral deposition [157]. Also, the continuity between cartilage and bone deserves excellent attention because of the biochemical and biomechanical cues between these two anatomical sites. Culture medium from osteoblasts undergoing cyclic compression may foster the release of MMP3 and MMP13 and reduce type II collagen and aggrecan in chondrocytes [188]. Recently, a sophisticated 3D model comprising a custom-built multi-well silicon loading plate carrying osteoblasts and osteocytes showed how this interplay reduces sclerostin (SOST) and promotes the release of NO and PGE_2_ [189].

Fluid shear stress is another typical force occurring after joint loading by the friction of the synovial fluid on the cartilage surface [171]. There is accumulating evidence from in vitro studies that low fluid shear exerts a chondroprotective function, whilst high fluid shear (>10 dyn/cm^2^) triggers several catabolic and inflammatory processes via the NF-κB signaling pathway [190]. Prolonged application of high fluid shear to human T/C-28a2 chondrocytes launches OA-related mechanisms in terms of the release of inflammatory mediators [191]. Shear stress is often used on articular chondrocytes to recapitulate OA-related mechanisms implicated in the production of inflammatory and catabolic mediators [190]. Moreover, shear stress also contributes to modulating the expression of tissue inhibitors of metalloproteinases (TIMPs) [192]. These results underline the relevant effects of physical forces on cells within joint tissues and how supraphysiological mechanical conditions trigger inflammatory changes. When referring to studies with tissue explants, we believe it is useful to perform macroscopic studies of the biological sample before collecting tissue explants. This preliminary analysis is necessary, as the biological sample often shows different histological features due to the regional and patchy distribution of OA. As for tissue-engineered constructs, we believe that dynamic conditions are better than static ones, as the first promote a higher synthesis of ECM [171]. Using engineered constructs ensures a better standardization than tissue explants for potential future applications in regenerative medicine (Table 2).

### 4.3. Evaluating Biomechanical OA Models in the Synovial Membrane 

The synovial cavity undergoes mechanical stress in both physiological and pathological conditions. In particular, the synovial membrane sustains load-bearing and shear forces derived from the synovial fluid flow. Several stress factors during OA, such as inflammation, can alter joint biomechanics with subsequent effects on all joint tissues, including the synovial membrane [192]. Yokota, H. and his coworkers evaluated the impact of impulsive mechanical loads on the release of proteolytic enzymes in two cell lines of synovial cells from healthy donors and patients with rheumatic arthritis. These forces can foster the synthesis of MMPs in the synovial cells, especially in cells from patients [193]. Similar studies have been performed by testing the modulation of mechanical loading on the anabolic and inflammatory mediators in synovial fibroblasts from healthy donors and OA patients. In particular, they proved that mechanical loading triggers the expression of several inflammatory mediators, such as TNF-α and PGE2 in both 2D and 3D synovial fibroblasts cultured on collagen scaffold [193,194,195]. Schroeder, A. and her colleagues showed the impact of biophysical cues on the extracellular matrix, in terms of an increase of type I collagen; highly expressed during OA [195]. Further shreds of evidence on the role exerted by cyclic compressive load were carried out on 3D synovial cells loaded onto a collagen scaffold. In particular, the authors gave evidence of high levels of MMP1, MMP3, MMP9, IL6, IL8, and IL1β following excessive compression [196] (Table 2). Similarly to cartilage, loading models contribute to modulating inflammatory and catabolic markers also in the synovial membrane, with the best results in 3D structures.

**Table 2 jcm-10-01920-t002:** List of some biomechanical models in OA with details of the kind of mechanical stimuli, the culture models, the main findings, and the bibliographic references.

Mechanical Stimuli	Culture Models	Main Results	Ref.
Cyclic compressive loading	3D culture: bovine chondrocytes in alginate beads	Induction of a hypertrophic phenotype through the release of RUNX 2 and collagen type X.	[171]
Single impact delivered from drop towers	Explants of articular cartilage	- Decreased proteoglycan biosynthesis with increasing impact stress; - Threshold stress of 15–20 MPa induced cell death and damage to the collagen network; - Cell death on the superficial zone of articular cartilage	[177]
IL-1β + osmotic stimulation	2D cultures of porcine articular chondrocytes	IL-1 interferes with the adaptation responses to mechanical loading by altering the Ca^2+^ response of chondrocytes with F-actin remodeling via small Rho GTPases	[178]
Drop tower for impaction	Osteochondral explants from the lateral tibial plateau	Enhanced phosphorylation of p38 and ERK1/2 in chondrocytes near impact sites following 24 h post-impaction.	[184]
Compression (single ramp compression, speed of 100%/s)	Bovine cartilage explants (size: 3 mm in diameter)	- Induction of compression through a custom incubator-housed loading apparatus; - Marked lubricin synthesis was a transient response following the injurious compression.	[185]
Dynamic compression (1 Hz) with 12 MPa stress	Articular cartilage explants from cow	Activation of the Smad 2/3P signaling pathway with a chondroprotective role by blocking hypertrophic differentiation	[186]
Physiological compression (1 Hz frequency/3 rounds of 20 min/20 h intervals)	Human OA cartilage explants (± IL-1β/IL4)	Tissue samples compressed with IL4 showed the best histological results (high collagen II, Sox-9, COMP, aggrecan).	[187]
Cyclic compression (40 KPa, 50 Hz)	Human SF on a collagen scaffold	Induction of an inflammatory phenotype by increasing the expression of PGE2, IL6, and IL-8.	[195]
Cyclic compressive load	Synovial cells on a collagen scaffold	Induction of MMP1, MMP3, MMP9, IL6, IL8, and IL1β.	[196]

### 4.4. Perspectives of Biomechanical OA Models in Regenerative Medicine: A Focus on MSCs

In vitro biomechanical models highlighted the impact of mechanical stimuli on tissue regeneration. Akin to joint cells, MSCs are mechano-sensitive to a greater extent than adult cells [167]. For this reason, many studies investigated the effects of biophysical cues on MSCs in the OA setting [171]. As reported for inflammatory models, the focus of these analyses was to explore the effect of mechanical stimuli on (i) MSCs migration; (ii) MSC differentiation, and (iii) MSC paracrine action.

#### 4.4.1. Biomechanical Models for Exploring MSCs Distribution

Several authors reported that biomechanical stimulation can modulate MSCs distribution. In particular, in vitro loading compression could promote MSCs homing from a reservoir to an alginate scaffold [194,195]. The feasibility of modulating cell migration through biophysical cues can have great biological relevance, especially in osteoarthritis tailored to attract progenitor cells towards the damaged joint tissues. However, it is mandatory to select the delivering systems of MSCs since it strongly influences how MSCs detect mechanical stimuli. 

#### 4.4.2. Biomechanical Models for Examining the Differentiation and Paracrine Profile of MSCs

The focus of several studies was to evaluate the impact of mechanical forces on the commitment of MSCs towards chondrogenic lineage [172,173,197]. Compression fosters chondrogenesis through the upregulation of GAG and collagen type II gene expression, reporting similar values to exogenous stimulation with TGFβ1 [172,198,199]. Further studies underlined that the loading starting time influences the mechano-responsiveness of MSCs to compression in the presence of TGFβ1 [199]. Combining different mechanical forces reflects more reliably what occurs in the in vivo OA microenvironment; thereby representing an attractive alternative for studying MSCs. Along this line, Cochis et al. investigated the influence of a combined model of compression and shear forces on the differentiation potential of MSCs embedded in a hydrogel. They observed optimal chondrogenesis of engineered constructs after 21 days of biophysical stimulation [173]; further studies confirmed these data [199]. Similar to bone marrow-derived MSCs, the differentiation potential of ASCs embedded in 3D porous polylactic-co-glycolic acid (PLGA) scaffold was also influenced by uniaxial dynamic compression (1 Hz) [174]. We believe that biomechanical models can be valid tools to test the effect of engineered constructs for cartilage repair under specific biophysical cues. Recently, several efforts have been addressed to investigate biophysical cues in 2D and 3D coculture systems. In this light, it was reported that sinusoidal dynamic stimulation (5%, 10%, and 15% strain amplitude, 0.5 Hz) improved chondrocyte differentiation in coculture systems with MSCs and chondrocytes [200]. Further researches tested the effect of coculturing ASCs with chondrocytes under cyclic compression (1 Hz, 10% to 40% strain, and 1 to 9 h/day stimulation duration) in bioreactors. This combined system can not only favor the expression of typical cartilaginous molecules but also suppress the expression of fibrotic and hypertrophic markers [201,202,203]. We believe that this model can be a valid tool to investigate the effect of several therapeutic strategies on typical OA hallmarks such as fibrosis and hypertrophy.

## 5. Microfluidics as Research Systems in OA: Their Pros and Cons in Regenerative Medicine

A significant challenge emerging among scientists is the demand for new in vitro research models capable of generating the organ complexity of OA. There is an urgent need for clinical mimicry in predictive preclinical studies. Microfluidic systems may be a promising alternative as they offer the great benefit of recreating dynamic flow conditions, biochemical, and mechanical stimuli in a closer way than classical culture methods. Thanks to 3D manufacturing, microfluidics may propose multitissue systems with nuanced and complex biological responses [20,204,205]. Along this line, Tuan, R.S. and his coworkers proposed an in vitro 3D model of the osteochondral unit by integrating a multichamber bioreactor in a microfluidic device. Beyond mimicking the osteochondral tissue with MSCs loaded onto collagen hydrogel, this system also has the significant advantage of simulating the synovial lining by using MSCs seeded onto polyethene glycol hydrogel. This multichamber device may represent a refined model to test the inflammatory responses exploiting the impact of several cytokines on osteochondral tissue, and synovial membrane [20]. However, this system shows some limits because of the lack of: (i) relationship with other joint tissues; (ii) immune-mediated reactions; and (iii) mechanical stimuli. More specific microfluidic systems exist when the research issue is addressed on testing the impact of biomechanics and the role of potential biological treatments. Barbero and his group have recently developed a load-induced OA on a chip by applying hyper-physiological (30%) compression. This cartilage-on-a-chip is a polydimethylsiloxane-based device, composed of human articular chondrocytes grown into 3D cell-embedded hydrogel, cultured with a chondrogenic medium, where it is possible to apply cyclic compression. The enormous potential of this system is the possibility of simulating the typical OA phenotype, including inflammation, hypertrophy, and catabolic events with the possibility of evaluating more reliably the potential of new therapies [206]. The need for recreating models capable of testing biological responses following simultaneous biochemical and mechanical alterations is one of the key challenges for scientists in forming a holistic view. In this light, Rosser, J. et al. proposed an integrated microfluidic system by matching the biochemical and mechanical OA features. This model has the enormous benefit of recapitulating nutrient gradients along with the cartilage thickness thanks to shear forces. This approach ensures close physiological cartilage behavior by providing a naturally low metabolism and cell placement. By exploiting these properties, some scientists examined the effect of triamcinolone, a typical treatment used as an intra-articular injection for relieving pain in OA patients. In this light, this system may provide enormous prospects for having preliminary data on the efficacy of injective treatments; with important perspectives in cartilage repair [207]. To gain more insights on the role of immune-mediated reactions in OA, Mondadori, C. et al. have recently proposed a microfluidic model simulating the monocyte extravasation from the bloodstream to the synovium [208]. This microfluidic chip contains two micro-chambers with chondrocytes and synovial fibroblasts in fibrin gel, split by a channel matching the synovial fluid. Indeed, this system provides a more accurate model of the synovial membrane compared to previous microfluidic models, as it reproduces both lining and sub-lining layers. This microfluidic model can be considered as an upgraded system in recreating the anatomical characteristics of the synovial membrane. However, it is still far from mimicking the “true” complexity of this tissue. Interestingly, it may be useful for testing first-line strategies targeting synovial membrane and especially the inflammatory macrophage population. Although this system shows several advantages, further improvements are still necessary to recreating typical immune-mediated reactions during OA. Despite the advances in this field, several issues still exist [205]. Manufacturing ECM has to consider the establishment of various structures, stiffness, flow rates, and cell sources mimicking joint architecture to develop a reliable multitissue organ (Table 3).

## 6. Conclusions and Future Research Outlook

The distance from animal models to clinical trials represents a bottleneck, with the risk of no clear-cut results because of species differences and causes such as ageing and sex difference [209]. In the last decade, the great emphasis on in vitro studies has come from new scientific and technological advances, which offer the enormous potential to predict tissue responses. Cells represent ideal building blocks to restore injured tissues; however, it is compulsory to place them in a 3D microenvironment for understanding their behavior in both healthy and damaged conditions. The shift from 2D to 3D in vitro inflammatory and mechanical cultures has been the first turning point, improving the biological relevance of in vitro studies to test regenerative strategies. Additive manufacturing methods allowed the development of 3D biomimetic and customized scaffolds that support cell viability and functions [210]. Among 3D printing approaches, electrospinning gained considerable attention as it generates structures with the submicron resolution by reproducing specific surface topography, which is another critical aspect to consider when studying cell behavior [211,212,213]. When referring to 3D models with scaffolds, their selection covers great importance, since findings vary greatly based on their nature and architecture. Indeed, bioreactors and microfluidics systems also contributed to improving current methodological approaches by introducing specific biochemical and mechanical issues in miniaturized systems [204,205,206,207,208]. Along this line, scientists designed and developed accurate in vitro approaches to model the multiple interactions among cells from joint tissues. Several authors have recently developed new structures for mimicking the osteochondral unit and proposed models for the extravasation of monocytes from the peripheral blood exploiting microfluidics [19,20,208]. However, several limitations still exist to improve in vitro reliable OA models. The major obstacle is to model the complexity of tissue and organ in a physiologically relevant way by creating the full diversity and crosstalk of all cell types, and inflammatory and biomechanical stimuli present in the articular joint. Although cocultures and microfluidics models explore critical biological aspects in OA, we believe that the feasibility of developing a joint-on-a-chip approach might represent a relevant scientific tool to recreate entirely the human OA joint, with enormous perspectives in regenerative medicine.

## Figures and Tables

**Table 3 jcm-10-01920-t003:** List of some recent microfluidic models in OA with details of the kind of stimuli, the microfluidic model, the main findings, and the bibliographic references.

Types of Stimuli	Microfluidics Model	Main Results	Ref.
**Inflammatory stimulus:** **+** **IL1β**	Multichamber bioreactor with: MSCs embedded in collagen scaffold for mimicking osteochondral unit; MSCs embedded in a polyethylene glycol hydrogel for simulating the synovial lining.	This system recreates the osteochondral microsystem by recreating the anatomical structures of cartilage and bone tissues; The great advantage of this system is the possibility of evaluating the interplay between cartilage and bone in OA; The disadvantage of this platform is the lack of a multitissues compartment within the articular joint.	[20]
**Mechanical stimulus** **hyperphysiological compression: 30%**	Polydimethylsiloxane-based device with human articular chondrocytes grown onto 3D cell-embedded hydrogel, cultured with chondrogenic medium.	This system promotes OA changes, including the launch of inflammation, and hypertrophic and catabolic events; This system could become a suitable tool for evaluating the effect of potential OA treatments on cartilage metabolism.	[206]
**Mechanical stimuli (shear forces) +** **inflammatory stimuli** **(IL1** **β** **+ TNF** **α** **)**	Multichamber containing 3D chondrocytes mimicking nutrient gradients thanks to the application of shear forces	This system allows establishment of the natural low metabolism and cell distribution in cartilage leading to cell differentiation. This system promotes hypertrophic and catabolic processes.	[207]
**Inflammatory stimuli:** **synovial fluid and monocyte** **infiltration**	The microfluidic model was composed of: two chambers including articular chondrocytes and synovial fibroblasts in fibrin gel; a channel matching the synovial fluid.	This platform could provide insights for testing the signaling pathways involved in monocyte recruitment.	[208]

## Data Availability

Not applicable.

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
