# Peer review of "A Roadmap of In Vitro Models in Osteoarthritis: A Focus on Their Biological Relevance in Regenerative Medicine"

_jcm, 2021, doi:10.3390/jcm10091920_

Round 1

Reviewer 1 Report

I have read this work with high interest. The only thing missing was the lack of attention and discussion of the problem with the availability of control groups for primary cultures isolated from patients - usually these are tissues after mechanical trauma, so the big disadvantage is the inability to obtain completely "clean tissues" or "not pro-inflammatory stimulation”. Please address this issue in the revision. 

The manuscript is easy to read and has an adequate continuity. I believe that the topic is well addressed when it comes to in vitro OA models - the authors described both models divided into different types of tissues (cartilage, synovium and subcartilage part of the bone), the arrangement of tissues in culture – 2D and 3D, the origin of the tissues (cell lines and primary cultures), as well as types of stimulation (chemical inflammatory agents [i.e. IL1b, TNFa etc.], synovial fluid from patients, or mechanical stimulation and even systems using bioreactors or cultures using the flow of media between the chambers. Each of the models mentioned is also described in terms of cell response to a given stimulation and how it relates to the clinical picture of OA.

Author Response

We thank the Reviewer for his/her kind suggestions and for the time spent in revising the manuscript, which improved the quality of the manuscript. We revised the manuscript according to the Reviewer's comments as reported below.

Revisions

I have read this work with high interest. The only thing missing was the lack of attention and discussion of the problem with the availability of control groups for primary cultures isolated from patients - usually these are tissues after mechanical trauma, so the big disadvantage is the inability to obtain completely "clean tissues" or "not pro-inflammatory stimulation”. Please address this issue in the revision. 

-) We thank and agree with the Reviewer for his/her insightful and valuable comments, which are a missing aspect in our review. We followed his/her suggestion by inserting and discussing this crucial aspect in the text. Most of the in vitro studies are carried out in specimens from OA patients who underwent joint prosthesis and who likely exhibit a basal impairment in tissue homeostasis. The inability of obtaining human healthy tissues devoid of inflammatory stimuli from surgical samples is a big issue to consider during the experimental design. In this context, several companies offer quality-controlled healthy cells from various tissue sources to help investigators to ensure control groups.

We added these comments within the text from line 260.

The manuscript is easy to read and has an adequate continuity. I believe that the topic is well addressed when it comes to in vitro OA models - the authors described both models divided into different types of tissues (cartilage, synovium and subcartilage part of the bone), the arrangement of tissues in culture – 2D and 3D, the origin of the tissues (cell lines and primary cultures), as well as types of stimulation (chemical inflammatory agents [i.e. IL1b, TNFa etc.], synovial fluid from patients, or mechanical stimulation and even systems using bioreactors or cultures using the flow of media between the chambers. Each of the models mentioned is also described in terms of cell response to a given stimulation and how it relates to the clinical picture of OA.

-) We thank the Reviewer for the time spent in revising our manuscript and his/her insightful and constructive comments, which contributed to improving the quality of the manuscript by discussing some missing aspects. We appreciate the Reviewer's comments; we tried to present a roadmap on the different in vitro OA models with their prospects for testing regenerative approaches. 

Reviewer 2 Report

In this work, authors have focused on the biological relevance of different in vitro models to recapitulate osteoarthritis-associated cellular effects. Specifically, in this work authors have focused on inflammatory and biomechanical models.

Authors have successfully summarized the state of the art of diverse in vitro model (2D and 3D) aimed to unravel the pathogenic mechanisms underlying osteoarthritis (OA) development in the context of inflammatory and mechanical stimuli.

A substantial portion of the data presented in this work is well known by researchers in the field. Nonetheless, this review would be very useful for the scientific community not only to order the information but also to easily identify the limitations of each different in vitro model.

The manuscript is well written and easy to follow for the readership. Moreover, it is focused on a hot topic field that will be of interest to researchers. Despite this, there are some minor changes that could be done in order to improve its quality.

Specific comments:

  • Headings from page 4: lines 163 and 198. “2D” is missing
  • Table 1: In the section of TNF and TGFB the information of the “culture methods” and “main results” seems that is fused. To avoid this each category should be correctly separated and identified by a different colour. It would be appreciated to perform the same modifications in all the tables of this work.
  • Although the paper is well written certain expression are used too many times like “In this light” and “Along this line”. It would be appreciated the change of some of these expressions to improve the reading of the text.

Author Response

We thank the Reviewer for his/her kind suggestions and for the time spent in revising the manuscript which improved the quality of the manuscript. We thank the Reviewer for his/her kind suggestions and for the time spent in revising the manuscript which improved the quality of the manuscript. We revised the manuscript according to the Reviewer's comments, as reported below.

Revisions

In this work, authors have focused on the biological relevance of different in vitro models to recapitulate osteoarthritis-associated cellular effects. Specifically, in this work authors have focused on inflammatory and biomechanical models. Authors have successfully summarized the state of the art of diverse in vitro model (2D and 3D) aimed to unravel the pathogenic mechanisms underlying osteoarthritis (OA) development in the context of inflammatory and mechanical stimuli. A substantial portion of the data presented in this work is well known by researchers in the field. Nonetheless, this review would be very useful for the scientific community not only to order the information but also to easily identify the limitations of each different in vitro model.

-) We thank the Reviewer for the time spent in revising our manuscript and his/her insightful and constructive comments, which contributed to improving the quality of the manuscript by discussing some missing aspects. We appreciate the Reviewer's comments; we tried to present a roadmap on the different in vitro OA models with their prospects for testing regenerative approaches. 

The manuscript is well written and easy to follow for the readership. Moreover, it is focused on a hot topic field that will be of interest to researchers. Despite this, there are some minor changes that could be done in order to improve its quality.

 -) We thank the Reviewer for his/her positive comments. We revised the manuscript according to the Reviewers suggestions to improve the quality of the manuscript.

Specific comments

Headings from page 4: lines 163 and 198. “2D” is missing

-) We thank the Reviewer for his/her comments and apologise for the mistakes. We inserted 2D in line 163 and 3D in line 198.

Table 1: In the section of TNF and TGFB the information of the “culture methods” and “main results” seems that is fused. To avoid this each category should be correctly separated and identified by a different colour. It would be appreciated to perform the same modifications in all the tables of this work.

-) We thank the Reviewer for his/her comments and apologise for the mistakes. We thank the Reviewer for his/her insightful suggestions in revising the tables using colours. We revised all the tables of this work to allow better comprehension.

Although the paper is well written certain expression are used too many times like “In this light” and “Along this line”. It would be appreciated the change of some of these expressions to improve the reading of the text.

-) We thank the Reviewer for his/her constructive suggestions to improve the reading of the text. We revised some terms and changed some of the certain expressions within the text. We used the “track changes” function in Microsoft Word to render visible all changes to Editors and Reviewers.

This manuscript is a resubmission of an earlier submission. The following is a list of the peer review reports and author responses from that submission.